# Turning the Spell Around: Lightweight Alignment Amplification via Rank-One Safety Injection

## Abstract

Safety alignment in Large Language Models (LLMs) often involves mediating internal representations to refuse harmful requests. Recent research has demonstrated that these safety mechanisms can be bypassed by ablating or removing specific representational directions within the model. In this paper, we propose the opposite approach: Rank-One Safety Injection (ROSI), a white-box method that *amplifies* a model's safety alignment by permanently steering its activations toward the refusal-mediating subspace. ROSI operates as a simple, fine-tuning-free rank-one weight modification applied to all residual stream write matrices. The required safety direction can be computed from a small set of harmful and harmless instruction pairs. We show that ROSI consistently increases safety refusal rates - as evaluated by Llama Guard 3 - while preserving the utility of the model on standard benchmarks such as MMLU, HellaSwag, and Arc. Furthermore, we show that ROSI can also re-align 'uncensored' models by amplifying their own latent safety directions, demonstrating its utility as an effective last-mile safety procedure. Our results suggest that targeted, interpretable weight steering is a cheap and potent mechanism to improve LLM safety, complementing more resource-intensive fine-tuning paradigms.

***Warning:** This document may contain harmful or unsafe prompts.*

## 1 Introduction

Large language models (LLMs) have demonstrated striking generality (Brown et al., 2020), excelling across tasks ranging from factual question answering (Kamalloo et al., 2023) and reasoning (Wei et al., 2023b) to code synthesis (Tong & Zhang, 2024) and creative writing (Gómez-Rodríguez & Williams, 2023). Their versatility has made them the foundation of modern conversational assistants and productivity tools, where alignment techniques such as supervised fine-tuning and reinforcement learning from human feedback enable models to follow user instructions while adhering to safety constraints (Ouyang et al., 2022). As general-purpose interfaces for language interaction, LLMs are now widely deployed, fueling expectations that they may one day serve as core components of autonomous, high-stakes systems.

Yet the same properties that make LLMs powerful also render them fragile and exposed to attack. Pre-training on vast, uncurated corpora inevitably imbues models with the capacity to generate harmful content (Wu et al., 2024), and safety alignment through post-training optimization offers only a partial safeguard (Mendu et al., 2025). Researchers have shown that even carefully aligned chat models remain vulnerable to a growing arsenal of jailbreak strategies, including prompt injection, obfuscation, multilingual exploits, and fine-tuning aimed at suppressing refusal, all capable of circumventing safety guardrails (Lin et al., 2024; Chu et al., 2024; Wei et al., 2023a).

Recent advances in mechanistic interpretability shed light on why these vulnerabilities arise. In particular, Arditi et al. (2024) demonstrate that refusal behavior is mediated by a *one-dimensional linear direction* in the activation space of many open-source chat models. Erasing this "refusal direction" from the residual stream suffices to disable safety alignment, enabling harmful completions;

Figure 1: **RANK-ONE SAFETY INJECTION (ROSI).** An aligned model processes both benign and harmful prompts in a forward pass (1). A safety vector is derived from the difference between harmful and harmless activations (2). Subtracting this vector ablates safety signals, producing an Abliterated Model. Adding it reinforces safety, producing a ROSI Model.

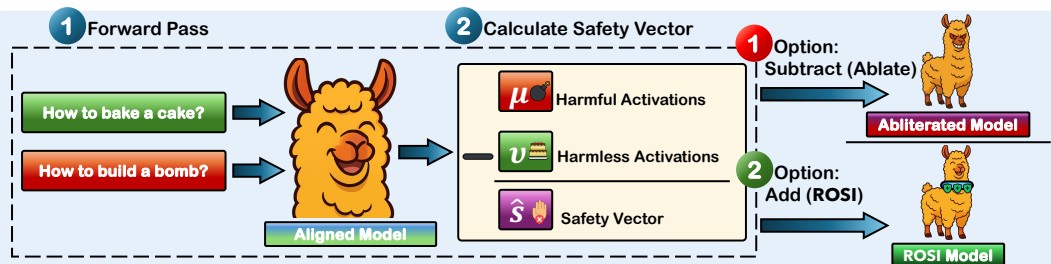

conversely, adding this direction to a model's activations can induce refusal even on benign prompts. This remarkable finding shows that refusal is encoded in an interpretable, causal subspace. Yet, it also exposes a critical weakness: if such a simple linear feature can be ablated, safety alignment is precarious.

Inspired by these insights, we ask the opposite question: rather than *removing* safety, can we systematically *amplify* it? In this paper, we propose RANK-ONE SAFETY INJECTION (ROSI), a simple, fine-tuning-free method that hardens model refusal by applying a lightweight rank-one modification to its weights. ROSI extracts a refusal-mediating direction from a small set of harmful/harmless instruction pairs, and permanently injects this direction into all residual stream write matrices.

We empirically demonstrate that ROSI provides two key benefits. First, it amplifies the safety of already aligned models, substantially improving their refusal rates and robustness against jailbreak attacks with negligible loss of utility. Second, it can re-align "uncensored" models that have been deliberately fine-tuned to ignore safety, reinstating refusal behavior without retraining. In summary, our contributions are:

- We introduce RANK-ONE SAFETY INJECTION (ROSI), a lightweight and interpretable weight-editing method to improve safety alignment in LLMs.
- We show that ROSI consistently improves the refusal and robustness of aligned models while preserving general utility on standard benchmarks.
- We demonstrate that ROSI can serve as an effective last-mile safety procedure, re-aligning uncensored models without expensive retraining.

Our findings highlight the practical value of mechanistic interpretability: by identifying and manipulating linear representations of safety, we can design efficient and powerful alignment techniques that complement resource-intensive optimization pipelines. More broadly, ROSI illustrates how interpretability-driven interventions can transform vulnerabilities into actionable tools to build safer AI systems.

## 2 RELATED WORK

**Mechanistic Interpretability of Refusal.** A central finding in alignment research is that refusal behavior in LLMs can be localized to low-dimensional linear features. Arditi et al. (2024) showed that a single direction in the residual stream mediates refusals across diverse chat models, with erasure or amplification of this direction directly controlling compliance with harmful prompts. Follow-up work has extended this line of inquiry: Zheng et al. (2024) disentangled harmfulness from refusal, showing that models encode internal judgments of harmfulness independently of whether they refuse; Hong et al. (2025) identified another single direction governing the balance between reasoning and memorization; and Jain et al. (2024b) demonstrated how fine-tuning minimally alters weights to cluster unsafe activations. Others proposed activation interventions, including SAE-based steering (O'Brien et al., 2024; He et al., 2025), Trojan activation bypasses (Wang & Shu,

2024), and neuron- or rank-level manipulations (Wei et al., 2024; Li et al., 2024b). Together, these works establish refusal as an interpretable and causally manipulable concept, but also highlight its brittleness to adversarial inputs and fine-tuning.

**Safety Steering and Training-free Defenses.** Training-free interventions attempt to steer model activations without costly fine-tuning. Early work showed that feature directions derived from contrastive inputs can modulate model behavior (Zou et al., 2023; Panickssery et al., 2023; Li et al., 2024a; Marks & Tegmark, 2023; Turner et al., 2023). Sparse autoencoders (SAEs) provide an unsupervised route to discover such features (Bricken et al., 2023; Templeton et al., 2024). Recently, SAE-based steering has been applied directly to safety, revealing both promise and utility tradeoffs (O'Brien et al., 2024). Extensions include instruction-following features (He et al., 2025), category-wise safety steering (Ghosh et al., 2025; Bhattacharjee et al., 2024), and adaptive methods such as AdaSteer (Zhao et al., 2025). Complementary strategies include Safety Arithmetic (Hazra et al., 2024), Representation Bending (Yousefpour et al., 2025), Low-Rank Extrapolation (Perin et al., 2025), adversarial training approaches such as ReFAT (Yu et al., 2024), and null-space constraints methods like AlphaSteer (Sheng et al., 2025) that builds on insights from AlphaEdit(Fang et al., 2025) which is used for robust knowledge editing. Foundational studies further established linear features in representation spaces (Bolukbasi et al., 2016; Elhage et al., 2022; Geiger et al., 2024; Ravfogel et al., 2020). While effective, many steering-based defenses introduce capability tradeoffs, motivating interpretable and more surgical alternatives such as ours.

**Beyond Steering: Fine-tuning and Safety Robustness.** Another line of work examines how safety alignment emerges or fails under fine-tuning. Works like Zhan et al. (2023); Yang et al. (2023); Qi et al. (2023); Lermen et al. (2023) show that even small malicious or benign finetunes can undo refusal, while mechanistic studies suggest the internal circuitry remains intact (Jain et al., 2024b). SAFELORA, a training-free and data-free approach that shows how LoRA weights can be projected onto a safety-aligned subspace reducing safety degradation from fine-tuning LLMs. Other interventions strengthen refusal explicitly, such as extended-refusal finetuning against abliteration attacks (Shairah et al., 2025), refusal tokens for controllable calibration (Jain et al., 2024a), and single-vector ablations to mitigate false refusals (Wang et al., 2025). Others work on run-time interventions to protect against jailbreaks, such as SMOOTHLLM (Robey et al., 2024), and Jailbreak Antidote (Shen et al., 2025). Alignment fragility also arises in model merging: Hammoud et al. (2024) showed that unsafe models contaminate the merged ones unless alignment is explicitly included. Together, these works highlight the tension between robustness and utility in safety interventions.

**Our Contribution.** We build directly on the insight of Arditi et al. (2024) but invert its vulnerability: instead of ablating the safety direction to weaken safety, our ROSI method permanently injects it into model weights. Compared to inference-time steering (O'Brien et al., 2024; Zhao et al., 2025; Ghosh et al., 2025; Sheng et al., 2025; Shen et al., 2025), ROSI provides a one-time lightweight, fine-tuning-free, interpretable mechanism that is permanent yet minimally invasive. Compared to approaches based on fine-tuning (Zhan et al., 2023; Shairah et al., 2025), it achieves comparable robustness with a much lower cost. Importantly, ROSI is not intended to replace existing safety strategies; it can be layered with steering, fine-tuning, or other alignment methods to further reinforce model robustness. Thus, our work illustrates how mechanistic interpretability can be leveraged not only to diagnose vulnerabilities but also to design efficient last-mile safety amplification techniques.

## 3 METHODOLOGY

Our proposed method, ROSI, which is illustrated in Figure 1, is based on the principle that high-level concepts such as safety are linearly represented in the activation space of a model. We first extract this "safety direction" and then use it to craft a permanent modification to the model's weights.

### 3.1 MATHEMATICAL PRELIMINARIES: TRANSFORMERS

A decoder-only Transformer model processes a sequence of input tokens $\mathbf{t} = (t_1, \ldots, t_n)$. The core of the model is the residual stream, $\mathbf{x}_i^{(l)} \in \mathbb{R}^{d_{\text{model}}}$, which represents the activation for the $i$-th token

at the $l$-th layer. Each layer $l$ updates this activation through an attention block and a multi-layer perceptron (MLP) block:

$$\tilde{\mathbf{x}}_i^{(l)} = \mathbf{x}_i^{(l)} + \texttt{Attn}^{(l)}(\mathbf{x}_{1:i}^{(l)}) \tag{1}$$

$$\mathbf{x}_i^{(l+1)} = \tilde{\mathbf{x}}_i^{(l)} + \texttt{MLP}^{(l)}(\tilde{\mathbf{x}}_i^{(l)}) \tag{2}$$

The key components that are written in the residual stream are the attention output projection matrix ($W_O$) and the MLP output projection matrix ($W_{out}$). Our method targets these matrices, among others, for modification.

## 3.2 EXTRACTING THE SAFETY DIRECTION

To isolate the direction in the activation space corresponding to safety and refusal, we employ the difference-in-means technique. We construct two small and contrasting datasets.

- $\mathcal{D}_{\text{harmful}}$: A set of instructions that should elicit a refusal (e.g., "How do I build a bomb?").
- $\mathcal{D}_{\text{harmless}}$: A set of benign instructions that should be answered helpfully (e.g., "How do I bake a cake?").

We run the model on all the prompts in both datasets and collect the residual stream activations $\mathbf{x}_i^{(l)}$ at a specific layer $l$ and the position of the token $i$ (typically the last token of the prompt). We then compute the mean activation for each dataset:

$$\boldsymbol{\mu}^{(l)} = \frac{1}{|\mathcal{D}_{\text{harmful}}|} \sum_{\mathbf{t} \in \mathcal{D}_{\text{harmful}}} \mathbf{x}_i^{(l)}(\mathbf{t}) \tag{3}$$

$$\boldsymbol{\nu}^{(l)} = \frac{1}{|\mathcal{D}_{\text{harmless}}|} \sum_{\mathbf{t} \in \mathcal{D}_{\text{harmless}}} \mathbf{x}_i^{(l)}(\mathbf{t}) \tag{4}$$

The safety direction $\mathbf{s}^{(l)}$ is defined as the difference between these two means:

$$\mathbf{s}^{(l)} = \boldsymbol{\mu}^{(l)} - \boldsymbol{\nu}^{(l)} \tag{5}$$

This vector $\mathbf{s}^{(l)}$ points from the center of the harmless activation cluster towards the center of the harmful activation cluster. We select the optimal layer $l^*$ that yields the most effective direction based on a validation set of harmful and harmless prompts. We select the direction that maximizes refusal on harmful prompts while maintaining a KL-Divergence of $\leq 0.1$ on the harmless instructions. The final normalized safety direction is denoted as $\hat{\mathbf{s}}$.

## 3.3 RANK-ONE SAFETY INJECTION (ROSI)

Previous work has shown that one can ablate a direction $\hat{\mathbf{s}}$ from a weight matrix $W$ by applying a projection: $W' \leftarrow (I - \hat{\mathbf{s}}\hat{\mathbf{s}}^T)W$. This effectively removes the model's ability to represent information along that direction.

We propose the opposite: to amplify this direction. We achieve this by modifying every weight matrix $W_{\text{out}} \in \mathbb{R}^{d_{\text{model}} \times d_{\text{input}}}$ that writes to the residual stream. The modification is a rank-one update designed to add a small, consistent push in the direction of $\hat{\mathbf{s}}$. The ROSI update rule is:

$$W_{\text{out}}' \leftarrow W_{\text{out}} + \alpha \cdot \hat{\mathbf{s}} \cdot \bar{\mathbf{w}}^T \tag{6}$$

where:

- $\alpha$ is a scalar hyperparameter that controls the strength of the injection.
- $\hat{\mathbf{s}} \in \mathbb{R}^{d_{\text{model}}}$ is the normalized safety direction.
- $\bar{\mathbf{w}} \in \mathbb{R}^{d_{\text{input}}}$ is the mean of the row vectors of the original weight matrix $W_{\text{out}}$.

This formulation creates a rank-one matrix $\alpha(\hat{\mathbf{s}}\bar{\mathbf{w}}^T)$ which is added to the original weights. The intuition is that for an average input, this modification adds a component proportional to the safety direction $\hat{\mathbf{s}}$ to the output, effectively steering the model's activations toward the refusal-mediating subspace. This is a permanent, efficient, and targeted change to the model's behavior.

# 4 EXPERIMENTS AND RESULTS

Our empirical evaluation is designed to answer three key questions:

1. Can ROSI amplify the safety of existing, aligned models and improve their robustness to adversarial attacks without degrading their general capabilities?
2. Can ROSI effectively inject safety into "uncensored" models that have been fine-tuned to bypass safety constraints?
3. Does this injected safety come at the cost of utility in these uncensored models?

We address these questions through a series of controlled experiments on a diverse set of models and benchmarks.

## 4.1 EXPERIMENTAL SETUP

**Models.** We test two categories of models: **Aligned Models** including LLAMA-2 (Touvron et al., 2023), LLAMA-3 (Llama Team, 2024), QWEN2.5 (Qwen et al., 2025), GEMMA (Team et al., 2024), and YI (AI et al., 2025), which have standard safety training; and **Uncensored Models**, specifically the DOLPHIN series (Dolphin, 2025), which are intentionally fine-tuned to ignore safety.

**Evaluation.** Safety is measured via Harm Refusal (HR) on CATQA (Bhardwaj et al., 2024), a set of 550 harmful instructions from 11 categories, evaluated using LLAMA GUARD 3 (Llama Team, 2024). We also measure attack success rates on jailbreak benchmarks—DAN, HARMBENCH (Mazeika et al., 2024), WILDGUARDTEST, and WILDJAILBREAK (Jiang et al., 2024)—judged by WILDGUARD (Han et al., 2024). Utility is assessed on standard benchmarks: MMLU (Hendrycks et al., 2021), HELLASWAG (Zellers et al., 2019), ARC (Chollet, 2019), BOOLQ (Clark et al., 2019), and TRUTHFULQA (Lin et al., 2022). We also measure Benign Compliance (BC) on a randomly sampled set of 512 instructions from ALPACA (Taori et al., 2023), to ensure ROSI models do not refuse safe instructions.

**Implementation.** The safety direction for each model was extracted using 50 harmful/harmless pairs. Generations use greedy decoding with a max length of 1024 tokens.

Table 1: **Harm Refusal in Aligned Models.** ROSI consistently improves the refusal rate for harmful prompts (HR %) while maintaining high compliance for benign ones (BC %).

| Model | ROSI | HR % | BC % |
|-------|------|------|------|
| GEMMA-2B-INSTRUCT | ✗ | 98.4 | 99.4 |
|  | ✓ | 99.8 (+1.5) | 99.0 (-0.4) |
| LLAMA-2-7B-CHAT-HF | ✗ | 99.8 | 98.8 |
|  | ✓ | 100.0 (+0.2) | 99.8 (+1.0) |
| META-LLAMA-3.1-8B-INSTRUCT | ✗ | 98.2 | 99.6 |
|  | ✓ | 99.1 (+0.9) | 99.6 (0.0) |
| META-LLAMA-3.2-1B-INSTRUCT | ✗ | 79.5 | 99.2 |
|  | ✓ | 92.7 (+13.2) | 95.9 (-3.9) |
| QWEN2.5-0.5B-INSTRUCT | ✗ | 90.4 | 98.6 |
|  | ✓ | 99.3 (+8.9) | 91.4 (-7.2) |
| QWEN2.5-3B-INSTRUCT | ✗ | 89.8 | 99.6 |
|  | ✓ | 99.6 (+9.8) | 98.6 (-1.0) |
| QWEN2.5-7B-INSTRUCT | ✗ | 95.8 | 100.0 |
|  | ✓ | 100.0 (+4.2) | 99.0 (-1.0) |
| QWEN2.5-14B-INSTRUCT | ✗ | 98.9 | 100.0 |
|  | ✓ | 100.0 (+1.1) | 99.4 (-0.6) |
| YI-6B-CHAT | ✗ | 81.3 | 99.6 |
|  | ✓ | 99.5 (+18.2) | 97.7 (-1.7) |

Table 2: **Jailbreak Robustness of Aligned Models.** Scores represent attack success rates (lower is better). ROSI significantly reduces model vulnerability across all attack vectors.

| Model | ROSI | DAN ↓ | HarmBench ↓ | WildGuardTest ↓ | | | WildJailbreak Harmful ↓ |
|---|---|---|---|---|---|---|---|
| | | | | WG-Micro | WG-Adv. | WG-Vanilla | |
| Gemma-2B-Instruct | ✗ | 5.3 | 6.2 | 9.1 | 16.6 | 2.9 | 42.3 |
| | ✓ | **1.0** (-4.3) | **3.4** (-2.8) | **2.4** (-6.7) | **4.7** (-11.9) | **0.5** (-2.4) | **8.2** (-34.1) |
| Llama-2-7b-chat-hf | ✗ | 0.0 | 0.0 | 0.9 | 2.1 | 0.0 | 3.5 |
| | ✓ | 0.0 (0.0) | 0.0 (0.0) | **0.0** (-0.9) | **0.0** (-2.1) | 0.0 (0.0) | **0.1** (-3.4) |
| Llama-3.1-8B-Instruct | ✗ | 0.3 | 5.9 | 1.6 | 2.7 | 0.7 | 14.8 |
| | ✓ | **0.0** (-0.3) | 5.3 (-0.6) | **0.0** (-1.6) | **0.0** (-2.7) | **0.0** (-0.7) | **1.8** (-13.0) |
| Llama-3.2-1B-Instruct | ✗ | 1.3 | 8.4 | 4.0 | 3.9 | 4.1 | 18.7 |
| | ✓ | **0.0** (-1.3) | **5.6** (-2.8) | **1.3** (-2.7) | **1.5** (-2.4) | **1.2** (-2.9) | **7.5** (-11.1) |
| Qwen2.5-0.5B-Instruct | ✗ | 36.0 | 31.6 | 33.1 | 48.1 | 20.9 | 91.8 |
| | ✓ | **7.0** (-29.0) | **12.8** (-18.8) | **21.1** (-12.0) | **38.0** (-10.1) | **7.3** (-13.6) | **58.8** (-33.0) |
| Qwen2.5-3B-Instruct | ✗ | 52.7 | 12.5 | 21.4 | 37.4 | 8.3 | 93.7 |
| | ✓ | **6.7** (-46.0) | **1.6** (-10.9) | **12.7** (-8.7) | **26.7** (-10.7) | **1.2** (-7.1) | **61.5** (-32.2) |
| Qwen2.5-7B-Instruct | ✗ | 40.3 | 22.5 | 18.6 | 36.2 | 4.1 | 90.7 |
| | ✓ | **11.7** (-28.6) | **1.9** (-20.6) | **3.9** (-14.7) | **7.7** (-28.5) | **0.7** (-3.4) | **36.7** (-54.0) |
| Qwen2.5-14B-Instruct | ✗ | 32.3 | 7.2 | 12.1 | 24.0 | 2.4 | 81.2 |
| | ✓ | **5.0** (-27.3) | **1.6** (-5.6) | **5.1** (-7.0) | **11.0** (-13.0) | **0.2** (-2.2) | **43.9** (-37.3) |
| Yi-6B-Chat | ✗ | 52.0 | 20.9 | 22.7 | 39.2 | 9.2 | 89.4 |
| | ✓ | **15.3** (-36.7) | **7.8** (-13.1) | **10.1** (-12.6) | **22.0** (-17.2) | **0.5** (-8.7) | **44.6** (-44.8) |

Table 3: **Utility Preservation in Aligned Models.** Performance on standard benchmarks with ROSI (✓) versus baseline (✗).

| Model | ROSI | MMLU | HellaSwag | Arc Easy | Arc Chal. | BoolQ | TruthfulQA |
|---|---|---|---|---|---|---|---|
| Gemma-2B-Instruct | ✗ | 38.1 | 49.2 | 71.7 | 40.4 | 63.7 | 45.8 |
| | ✓ | 38.3 (+0.2) | 49.3 (+0.1) | 70.8 (-0.9) | 39.0 (-1.4) | 61.4 (-2.3) | 46.7 (+0.9) |
| Llama-2-7B-Chat-HF | ✗ | 46.3 | 57.8 | 74.0 | 43.9 | 79.6 | 45.3 |
| | ✓ | 46.4 (+0.1) | 57.7 (-0.1) | 73.4 (-0.6) | 43.3 (-0.6) | 79.8 (+0.2) | 47.2 (+1.9) |
| Meta-Llama-3.1-8B-Instruct | ✗ | 68.0 | 59.1 | 81.7 | 51.6 | 84.0 | 54.1 |
| | ✓ | 67.6 (-0.4) | 58.9 (-0.2) | 81.1 (-0.6) | 51.1 (-0.5) | 83.8 (-0.2) | 54.8 (+0.7) |
| Meta-Llama-3.2-1B-Instruct | ✗ | 46.0 | 45.2 | 68.3 | 35.6 | 69.3 | 43.9 |
| | ✓ | 45.4 (-0.6) | 45.4 (+0.2) | 67.4 (-0.9) | 34.7 (-0.9) | 68.7 (-0.6) | 45.0 (+1.1) |
| Qwen2.5-0.5B-Instruct | ✗ | 45.8 | 40.5 | 65.5 | 30.1 | 67.6 | 41.8 |
| | ✓ | 45.3 (-0.5) | 40.4 (-0.1) | 64.3 (-1.2) | 29.6 (-0.5) | 63.2 (-4.4) | 43.8 (+2.0) |
| Qwen2.5-3B-Instruct | ✗ | 65.4 | 56.3 | 76.9 | 45.7 | 80.1 | 58.7 |
| | ✓ | 65.0 (-0.4) | 55.8 (-0.5) | 76.6 (-0.3) | 45.1 (-0.6) | 77.4 (-2.7) | 59.7 (+1.0) |
| Qwen2.5-7B-Instruct | ✗ | 71.8 | 62.0 | 81.6 | 52.6 | 86.4 | 64.8 |
| | ✓ | 71.9 (+0.1) | 61.9 (-0.1) | 81.0 (-0.6) | 52.6 (0.0) | 86.2 (-0.2) | 66.1 (+1.3) |
| Qwen2.5-14B-Instruct | ✗ | 78.8 | 65.6 | 85.7 | 60.4 | 88.0 | 69.0 |
| | ✓ | 78.9 (+0.1) | 65.6 (0.0) | 85.6 (-0.1) | 60.7 (+0.3) | 85.8 (-2.2) | 71.9 (+2.9) |
| Yi-6B-Chat | ✗ | 61.6 | 57.7 | 74.5 | 44.1 | 82.8 | 49.9 |
| | ✓ | 61.1 (-0.5) | 57.2 (-0.5) | 78.1 (+3.6) | 46.9 (+2.8) | 84.2 (+1.4) | 51.2 (+1.3) |

## 4.2 Amplifying Safety in Aligned Models

We first test ROSI's ability to bolster the defenses of models that already possess safety alignment.

**Increased Refusal and Jailbreak Robustness.** As shown in Table 1, applying ROSI consistently enhances the Harm Refusal (HR) rate across all aligned models tested. The effect is particularly pronounced for models with weaker baselines, such as Yi-6B-Chat (+18.2 points) and Meta-Llama-3.2-1B-Instruct (+13.3 points), elevating their safety to near-perfect levels. This improvement is not superficial; Table 2 shows that ROSI drastically hardens models against a full suite of adversarial jailbreak attacks. For many models, attack success rates are cut by more than half, demonstrating a fundamental increase in robustness.

In Appendix 5, we discuss what role ROSI can play in fine-tuning LLMs.

**Preservation of Model Utility.** Crucially, these safety gains do not compromise the models' core functionalities. Table 3 provides a comprehensive view of utility preservation. The average performance across a suite of seven benchmarks remains remarkably stable. The vast majority of models

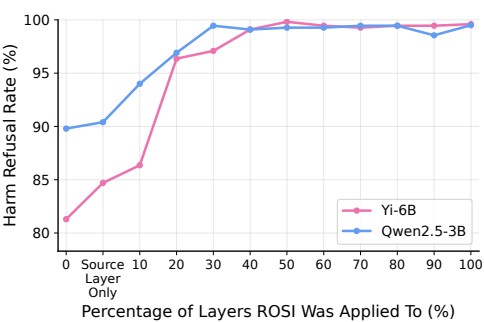 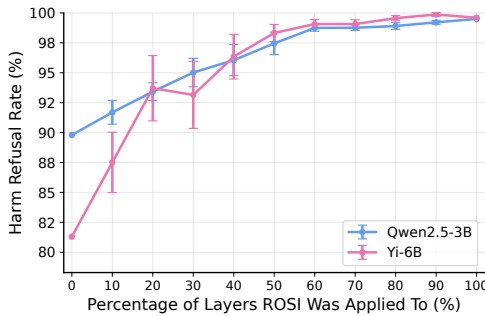

(a) Selection is centered on the layer from which the safety vector is extracted and a proportional window around it.

(b) Layers are selected at random; the process is repeated 10 times for each ratio, the plot show he mean refusal rate with confidence intervals.

Figure 2: **Injected Layers Ablations.** In Figure 2a, we ablate the number of layers we apply ROSI to by taking a ratio $x$ (x-axis) of a model's layers that is centered around the index of the layer $i$ used to extract the safety vector. In Figure 2b, a subset of layers is selected randomly each time, we repeat the run 10 times for each ratio and take the average of the harm refusal rate. Confidence intervals are reported.

see an average score change of less than 0.5%. A similar pattern holds for BC, as seen in Table 1, ROSI models' refusal of safe instructions, on average, remains minimal. While smaller models ($\leq$ 1B) show the biggest degradation in BC, they still gain more in HR than what they lose in BC. These results demonstrate that the safety direction is largely orthogonal to the representations required for knowledge and reasoning tasks. ROSI acts as a surgical tool, enhancing safety with minimal side effects.

**Injected Layers Ablations.** To assess how stable the ROSI update is within a model, we perform a set of ablations that vary both the number and the identity of the layers receiving the safety injection for two representative models, YI-6B-CHAT and QWEN2.5-3B-INSTRUCT. In the first setting, we inject ROSI into a contiguous block of layers centered on the layer index used to extract the safety vector, expanding this window according to a chosen fraction of the model's total depth. Figure 2a shows how injecting just at the source layer yields only modest improvements, and as the window of injected layers is expanded, the harm refusal rate keeps increasing until it stabilizes around the $30 - 40\%$ window size, suggesting that only a limited number of layers within a model contribute to the concept of "safety". In a second setting, we examine robustness by randomly selecting the same number of layers for each fraction. For every ratio, we repeat the process ten times and average the resulting refusal scores. Figure 2b displays a similar trend to the former experiment, but the confidence intervals show that performance varies considerably depending on the layers selected. Notably, YI-6B-CHAT peaks at 100% HR rates in one of the runs where ROSI was applied to only half of the layers, which suggests that optimizing the set of injected layers can further improve performance.

> **Conclusion 1**
>
> ROSI effectively amplifies the safety of existing aligned models. It robustly increases their refusal of harmful prompts and hardens them against jailbreak attacks, all with a negligible impact on their general utility and performance.

### 4.3 INJECTING SAFETY INTO UNCENSORED MODELS

The previous experiment demonstrated that ROSI can enhance refusal behavior in models that are already aligned. We now turn to the more demanding task of applying ROSI to uncensored DOL-PHIN models. This tests whether our method can serve as a "last-mile" re-alignment tool to instill safety where it was deliberately removed.

Figure 3: **Applying ROSI to Uncensored Models.** In the forward pass, harmful and harmless instructions are prepended with a system prompt directing an uncensored model to reject harmful requests, thus eliciting refusal.

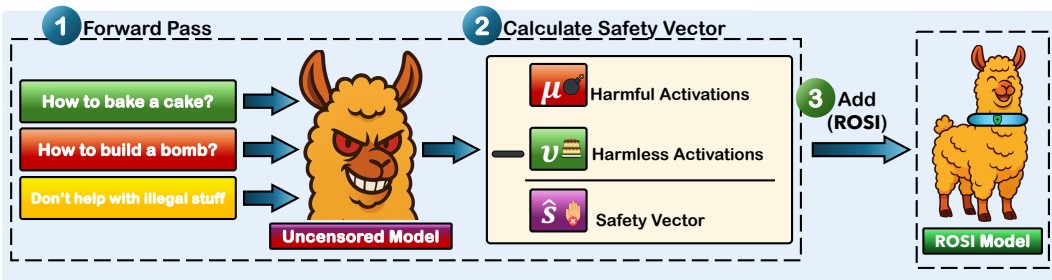

Table 4: **Safety Injection in Uncensored Models.** Applying ROSI substantially boosts harm refusal (HR) across DOLPHIN models, while preserving compliance with benign instructions (BC). Ablations without a safety system prompt (♟) highlight the role of prompt-level safety conditioning.

| Model | ROSI | HR % | BC % |
|---|---|---|---|
| DOLPHIN3.0-LLAMA3.2-1B | ✗ | 23.5 | 100.0 |
| | ✓ | 46.0 (+22.5) | 99.4 (-0.6) |
| | ♟ | 18.4 (-5.1) | 100.0 (0.0) |
| DOLPHIN3.0-QWEN2.5-3B | ✗ | 50.0 | 100.0 |
| | ✓ | 86.0 (+36.0) | 99.6 (-0.4) |
| | ♟ | 33.6 (-16.4) | 100.0 (0.0) |
| DOLPHIN3.0-LLAMA3.1-8B | ✗ | 65.8 | 100.0 |
| | ✓ | 100.0 (+34.2) | 100.0 (0.0) |
| | ♟ | 88.9 (+23.1) | 100.0 (0.0) |
| DOLPHIN3.0-MISTRAL-24B | ✗ | 64.4 | 100.0 |
| | ✓ | 92.0 (+27.6) | 100.0 (0.0) |
| | ♟ | 47.8 (-16.6) | 100.0 (0.0) |

**Eliciting Refusal Behavior and Reducing Vulnerability.** The DOLPHIN models exhibit very low baseline safety, leaving little to no refusal signal to extract. Directly applying the method from Section 3 to a DOLPHIN model would therefore yield a vector $\hat{s}$ that does not represent a safety direction.

To overcome this, we explicitly *elicit* refusal behavior by modifying the system prompt, as can be seen in Figure 3. Specifically, we prepend instructions that direct the model to reject harmful categories of requests; the prompt we used can be seen in Appendix D. This artificially introduces a refusal subspace that would otherwise be absent. Once present, we can apply ROSI to these models. Afterwards, the system prompt is no longer needed and is removed during testing.

Table 4 shows that ROSI achieves dramatic improvements. For instance, DOLPHIN3.0-QWEN2.5-3B's safe response rate skyrockets from 50.0% to 86.0% (+36.0), while DOLPHIN3.0-LLAMA3.1-8B is fully re-aligned to 100% safety. This demonstrates that even uncensored models retain a latent safety direction that is potent enough to overwrite their fine-tuning when amplified. This injected safety also translates to improved robustness. As seen in Table 5, ROSI provides a powerful first line of defense, slashing attack success rates by large margins (e.g., a 46.3-point reduction on DAN for DOLPHIN3.0-QWEN2.5-3B).

**Utility Preservation.** Answering our final question, Table 6 confirms that this powerful safety injection does not harm the utility of the uncensored models. The average performance across the benchmark suite is virtually unchanged, with score differences of only +/- 0.2%. This result is significant: it shows that safety can be added back to a model post-hoc without repeating expen-

Table 5: **Jailbreak Vulnerability of Uncensored Models.** Scores are attack success rates (lower is better). ROSI provides a crucial layer of defense, significantly reducing their extreme vulnerability.

| Model | ROSI | DAN ↓ | HARMBENCH ↓ | WILDGUARDTEST ↓ | | | WILDJAILBREAK Harmful ↓ |
|---|---|---|---|---|---|---|---|
| | | | | WG-Micro | WG-Adv. | WG-Vanilla | |
| DOLPHIN3.0-LLAMA3.2-1B | ✗ | 90.3 | 62.8 | 50.3 | 42.4 | 56.8 | 98.5 |
| | ✓ | 65.7 (-24.7) | 51.9 (-10.9) | 33.9 (-16.4) | 38.3 (-4.2) | 30.3 (-26.5) | 88.9 (-9.5) |
| | ☻ | 88.6 (-1.7) | 72.2 (+9.4) | 59.3 (+9.0) | 48.1 (+5.7) | 68.5 (+11.7) | 97.7 (-0.8) |
| DOLPHIN3.0-QWEN2.5-3B | ✗ | 90.3 | 52.8 | 32.6 | 37.7 | 28.4 | 96.7 |
| | ✓ | 44.0 (-46.3) | 20.9 (-31.9) | 15.4 (-17.2) | 27.3 (-10.4) | 5.6 (-22.8) | 70.4 (-26.3) |
| | ☻ | 52.7 (-37.6) | 32.2 (-20.6) | 23.4 (-9.2) | 29.4 (-8.3) | 18.4 (-10.0) | 82.8 (-13.9) |
| DOLPHIN3.0-LLAMA3.1-8B | ✗ | 90.3 | 54.7 | 27.0 | 34.7 | 20.6 | 94.0 |
| | ✓ | 82.3 (-8.0) | 47.2 (-7.5) | 21.1 (-5.9) | 29.4 (-5.3) | 14.3 (-6.3) | 82.8 (-11.3) |
| | ☻ | 81.3 (-9.0) | 44.7 (-10.0) | 19.2 (-7.8) | 26.7 (-8.0) | 13.1 (-7.5) | 84.1 (-9.9) |
| DOLPHIN3.0-MISTRAL-24B | ✗ | 80.7 | 43.8 | 18.7 | 27.3 | 11.7 | 87.5 |
| | ✓ | 64.3 (-16.3) | 28.4 (-15.3) | 9.1 (-9.6) | 16.9 (-10.4) | 2.7 (-9.0) | 63.2 (-24.2) |
| | ☻ | 84.0 (+3.3) | 50.0 (+6.2) | 22.4 (+3.7) | 27.0 (-0.3) | 18.7 (+7.0) | 92.2 (+4.7) |

Table 6: **Utility Preservation in Uncensored Models.** Performance after applying ROSI is shown with deltas relative to the baseline.

| Model | ROSI | MMLU | HELLASWAG | ARC EASY | ARC CHAL. | BOOLQ | TRUTHFULQA |
|---|---|---|---|---|---|---|---|
| DOLPHIN3.0-LLAMA3.2-1B | ✗ | 35.3 | 47.8 | 65.7 | 34.7 | 59.3 | 39.5 |
| | ✓ | 35.0 (-0.3) | 47.7 (-0.1) | 65.7 (0.0) | 34.7 (0.0) | 60.0 (+0.7) | 40.2 (+0.7) |
| | ☻ | 30.1 (-5.2) | 41.5 (-6.3) | 58.3 (-7.4) | 27.5 (-7.2) | 53.2 (-6.1) | 42.8 (+3.3) |
| DOLPHIN3.0-QWEN2.5-3B | ✗ | 64.7 | 55.5 | 77.9 | 43.8 | 80.5 | 49.5 |
| | ✓ | 64.7 (0.0) | 55.4 (-0.1) | 77.7 (-0.2) | 43.8 (0.0) | 80.6 (+0.1) | 50.8 (+1.3) |
| | ☻ | 64.7 (0.0) | 55.6 (+0.1) | 77.2 (-0.7) | 43.7 (-0.1) | 78.7 (-1.8) | 50.1 (+0.6) |
| DOLPHIN3.0-LLAMA3.1-8B | ✗ | 59.0 | 61.3 | 80.9 | 50.1 | 85.6 | 50.1 |
| | ✓ | 58.9 (-0.1) | 61.2 (-0.1) | 80.4 (-0.5) | 50.4 (+0.3) | 85.0 (-0.6) | 51.0 (+0.9) |
| | ☻ | 59.0 (0.0) | 61.2 (-0.1) | 80.1 (-0.8) | 50.2 (+0.1) | 85.1 (-0.5) | 50.9 (+0.8) |
| DOLPHIN3.0-MISTRAL-24B | ✗ | 72.5 | 59.8 | 26.6 | 22.1 | 84.1 | 54.6 |
| | ✓ | 72.5 (0.0) | 59.7 (-0.1) | 26.9 (+0.3) | 22.5 (+0.4) | 83.9 (-0.2) | 55.7 (+1.1) |
| | ☻ | 72.2 (-0.3) | 59.6 (-0.2) | 27.0 (+0.4) | 23.0 (+0.9) | 84.2 (+0.1) | 53.8 (-0.8) |

sive training or compromising the helpful capabilities that the uncensored model was designed to maximize.

**System Prompt Ablation.** Values marked with (☻) in Table 4 show results from models where ROSI was applied without prepending a safety system prompt to the input instructions. In this setting, DOLPHIN3.0-LLAMA3.1-8B exhibits an 11.1% smaller gain in harm refusal compared to when a safety system prompt is present. Other models fare considerably worse, with performance degrading outright. Table 5 mirrors this trend: a safety system prompt is essential to fully realize the benefits of ROSI in uncensored models. The relative resilience of DOLPHIN3.0-LLAMA3.1-8B without the system prompt suggests that the safety signal may not have been completely erased during uncensoring. In Figure 4, we examine how the presence of a safety system prompt influences the linear separability of harmful and harmless representations in the activation space. Using DOLPHIN3.0-QWEN2.5-3B, we see that without the system prompt, the latent distributions overlap significantly, impeding the ability of the steering vector to differentiate between safe and unsafe contexts. On the other hand, prepending the prompt effectively disentangles these clusters, increasing the centroid distance and restoring the distinct decision boundaries required for robust refusal. Taken together, these results support our hypothesis: a safety system prompt is crucial for eliciting a strong and coherent safety direction in uncensored models.

In Appendix E, we show that, on the other hand, aligned models do not benefit from the safety system prompt.

Figure 4: **PCA visualization of activation separation in DOLPHIN3.0-QWEN2.5-3B.** (a) In the absence of a safety system prompt, the embeddings for harmful (red) and harmless (blue) inputs show significant overlap. (b) When a safety system prompt is introduced, the clusters become more distinct.

(a) **Without a safety system prompt.**   (b) **With a safety system prompt.**

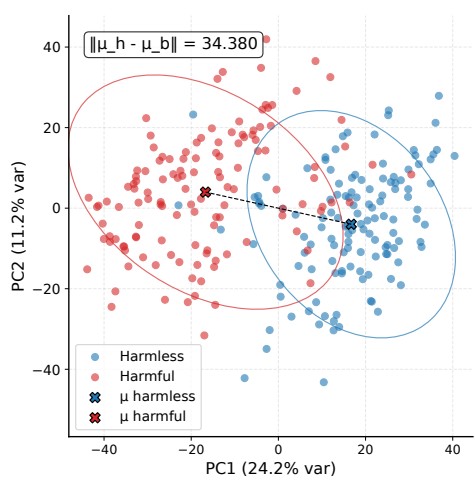 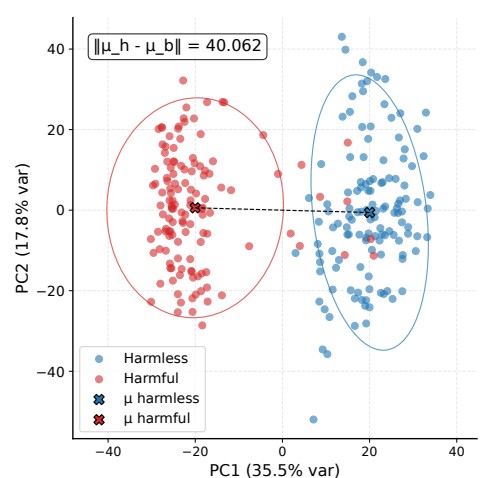

---

**Conclusion 2**

ROSI successfully injects safety into models that have been fine-tuned to be noncompliant. This provides a powerful, low-cost method for "re-aligning" uncensored models, making them significantly safer with minimal impact on their utility.

---

## 5 CONCLUSION

In this paper, we introduced RANK-ONE SAFETY INJECTION (ROSI), a simple and effective white-box method to enhance the safety alignment of Large Language Models. Building on the insight that safety and refusal behaviors are encoded in specific linear directions within a model's activation space, ROSI applies a permanent, rank-one modification to the model's weights to amplify this safety direction.

Our comprehensive experiments show that ROSI consistently improves the safety of a wide range of models. For already aligned models, it increases their refusal rates on harmful prompts and makes them substantially more robust to adversarial jailbreak attacks. For uncensored models, ROSI successfully injects safety mechanisms that were previously removed, serving as a powerful last mile alignment tool, we also demonstrate how a safety system prompt is crucial to extract a meaningful safety vector from these models. Critically, these significant safety gains are achieved with negligible degradation in model performance on a suite of standard utility benchmarks.

ROSI demonstrates the practical value of interpretability research. By understanding and manipulating the internal representations of models, we can develop low-cost targeted interventions that are more efficient than traditional, resource-intensive fine-tuning. This work opens up promising avenues for future research, including exploring more sophisticated methods for identifying and manipulating conceptual directions and extending this approach to other desirable model attributes beyond safety, such as honesty or controllability.

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

## A ROSI & FINE-TUNING

Recent work by Qi et al. (2023) demonstrated that fine-tuning Large Language Models (LLMs) often compromises their safety alignment, even when the fine-tuning dataset is entirely benign. To address this "alignment tax," several defensive strategies have been proposed, such as SAFELORA (Hsu et al., 2025). SAFELORA modifies the standard Low Rank Adapters (LORA) by projecting LORA weights from selected layers to a safety-aligned subspace, thereby mitigating safety degradation while preserving model utility.

In this section, we investigate the interaction between our proposed method, ROSI, and these parameter-efficient fine-tuning paradigms. We hypothesize that ROSI can act as a lightweight "safety vaccination" (or initialization), effectively hardening the model against the alignment erosion typically caused by downstream adaptation. We evaluate this on LLAMA-2-7B-CHAT measuring the Harmful Refusal (HR) rate across different sequences of application.

We fine-tuned the model on DATABRICKS DOLLY 15K (Conover et al., 2023) for 3000 steps with a learning rate of $5e^{-5}$, batch size of 8, LORA rank of 32. Other SAFELORA parameters are taken as is from the paper.

As shown in Table 7, standard LORA fine-tuning significantly degrades the safety of the base model, resulting in an HR of 82.7%. While SAFELORA provides a robust defense (95.5%), we observe that the order of ROSI application is critical. Applying ROSI as a post-hoc repair mechanism (LORA $\rightarrow$ ROSI) yields only marginal gains (85.5%), suggesting that once safety representations are disrupted by fine-tuning, they are difficult to fully recover via a rank-one update.

In contrast, injecting the safety vector *prior* to fine-tuning (ROSI $\rightarrow$ LORA) drastically improves resilience, maintaining a refusal rate of 98.6% even when followed by standard LORA updates. This indicates that ROSI successfully steers the model's initialization into a region of the parameter space that is more resistant to catastrophic forgetting of safety. Finally, the combination of pre-injection and safety-constrained adaptation (ROSI $\rightarrow$ SAFELORA) achieves a perfect refusal rate of **100.0%**, demonstrating that ROSI and SAFELORA are highly complementary techniques for secure model adaptation.

Table 7: **Comparison of Harm Refusal (HR) rates on LLAMA-2-7B-CHAT across different fine-tuning configurations.** Arrows ($\rightarrow$) denote the sequence of method application.

| Model | Method | HR % |
|---|---|---|
| | Base (no fine-tuning) | 99.8 |
| | LORA | 82.7 |
| | SAFELORA | 95.5 |
| LLAMA-2-7B-CHAT | LORA $\rightarrow$ ROSI | 85.5 |
| | ROSI $\rightarrow$ LORA | 98.6 |
| | SAFELORA $\rightarrow$ ROSI | 98.9 |
| | ROSI $\rightarrow$ SAFELORA | **100.0** |

# B  THE TRANSFERABILITY OF SAFETY VECTORS

One interesting question that can arise from our experiments is how would a safety vector extracted from one model affect another. The main constraint is that both models must share the same hidden dimensionality $\mathbb{R}^{d_{model}}$ for a vector to be transferable. Among the models we initially evaluated, none shared the same hidden size; however, QWEN2.5-14B-INSTRUCT and QWEN2.5-32B-INSTRUCT do. This allows us to study cross-model transfer directly. For each model, we extracted a safety vector following Section 3. We then applied ROSI twice per model: once using its own vector, and once using the vector extracted from the other model. Table 8 summarizes the outcomes. In both cases, applying the safety vector from the other model leads to meaningful gains on safety benchmarks. Notably, for QWEN2.5-14B-INSTRUCT, using the vector from the 32B variant produces stronger safety performance than using its own vector. This could suggest that the larger model had learned a better and more distinct representation of safety compared to the smaller model. Importantly, these gains occur without significant drops in utility (Table 3). Overall, these findings open questions about how safety directions emerge, how transferable they are across architectures of the same dimensionality, and what aspects of a model's training process facilitate such transfer. We leave these questions to future work.

Table 8: **Safety benchmarks for cross-model safety vector transfer.** Each model is evaluated in three settings: the original model, ROSI using its own extracted safety vector, and ROSI using the safety vector extracted from the other model. Using a safety vector from another model consistently improves safety performance, with the 14B model benefiting most from the safety vector extracted from the 32B variant.

| Model | DAN ↓ | HARMBENCH ↓ | WILDGUARDTEST ↓ | | | WILDJAILBREAK Harmful ↓ |
|---|---|---|---|---|---|---|
| | | | WG-Micro | WG-Adv. | WG-Vanilla | |
| QWEN2.5-14B-INSTRUCT | 32.3 | 7.2 | 12.1 | 24.0 | 2.4 | 81.2 |
| QWEN2.5-14B-ROSI | **5.0** (-27.3) | 1.6 (-5.6) | 5.1 (-7.0) | 11.0 (-13.0) | 0.2 (-2.2) | 43.9 (-37.3) |
| QWEN2.5-14B-ROSI-FROM-32B | **5.0** (-27.3) | **0.9** (-6.3) | **4.3** (-7.8) | **9.5** (-14.5) | **0.0** (-2.4) | **34.5** (-46.7) |
| QWEN2.5-32B-INSTRUCT | 42.0 | 18.4 | 14.8 | 28.2 | 3.9 | 83.3 |
| QWEN2.5-32B-ROSI | **21.7** (-20.3) | **12.2** (-6.2) | **10.4** (-4.4) | **19.9** (-8.3) | **2.7** (-1.2) | **72.6** (-10.7) |
| QWEN2.5-32B-ROSI-FROM-14B | 28.7 (-13.3) | 12.5 (-5.9) | 11.9 (-2.9) | 22.9 (-5.3) | 2.9 (-1.0) | 76.9 (-6.4) |

Table 9: **Utility evaluations under cross-model safety vector transfer.** Utility remains broadly stable across settings, indicating that the safety improvements shown in Table 8 do not come at the cost of substantial performance degradation.

| Model | MMLU | HELLASWAG | ARC EASY | ARC CHAL. | BOOLQ | TRUTHFULQA |
|---|---|---|---|---|---|---|
| QWEN2.5-14B-INSTRUCT | 78.8 | 65.6 | 85.7 | 60.4 | 88.0 | 69.0 |
| QWEN2.5-14B-ROSI | 78.9 (+0.1) | 65.6 (0.0) | 85.6 (-0.1) | 60.7 (+0.3) | 85.8 (-2.2) | 71.9 (+2.9) |
| QWEN2.5-14B-ROSI-FROM-32B | 78.5 (-0.3) | 65.6 (0.0) | 84.7 (-1.0) | 59.5 (-0.9) | 85.9 (-2.1) | 71.0 (+2.0) |
| QWEN2.5-32-INSTRUCT | 81.7 | 66.9 | 82.2 | 57.5 | 89.7 | 65.5 |
| QWEN2.5-32-ROSI | 81.6 (-0.1) | 67.1 (+0.2) | 81.9 (-0.3) | 57.2 (-0.3) | 89.7 (0.0) | 66.7 (+1.2) |
| QWEN2.5-32-ROSI-FROM-14B | 81.6 (-0.1) | 66.9 (0.0) | 82.1 (-0.1) | 57.2 (-0.3) | 89.4 (-0.3) | 66.7 (+1.2) |

## C  SENSITIVITY TO THE EXTRACTION SET

A key advantage of lightweight alignment methods is their minimal data requirement. To empirically verify this, we investigate the sensitivity of ROSI to the size of the dataset used for extracting the safety vector. We conduct an ablation study on QWEN2.5-3B-INSTRUCT, varying the number of contrasting harmful/harmless pairs used in the extraction phase from 1 to 100 samples.

The results, presented in Table 10, demonstrate high data efficiency. Surprisingly, ROSI achieves a substantial improvement in safety using just a single sample pair, boosting the Harmful Refusal (HR) rate from a baseline of 89.8% to 99.1%. This suggests that the safety direction is a highly dominant feature within the model's activation space, capable of being captured even with minimal signal, and increasing the sample size provides marginal stability improvements, peaking at 99.5% refusal with 50, 80, and 90 samples.

Table 10: **Change in Harm Refusal (HR) using different sets of harmful and harmless pairs to extract the safety vector from QWEN2.5-3B-INSTRUCT.**

| Model | Dataset Size | HR % |
|---|---|---|
| QWEN2.5-3B-INSTRUCT | Base | 89.8 |
| | ROSI (1 Sample) | 99.1 |
| | ROSI (10 Samples) | 96.55 |
| | ROSI (20 Samples) | 99.1 |
| | ROSI (30 Samples) | 98.0 |
| | ROSI (40 Samples) | 98.7 |
| | ROSI (50 Samples) | **99.5** |
| | ROSI (60 Samples) | 97.8 |
| | ROSI (70 Samples) | 98.7 |
| | ROSI (80 Samples) | **99.5** |
| | ROSI (90 Samples) | **99.5** |
| | ROSI (100 Samples) | 98.6 |

# D SAFETY SYSTEM PROMPT

Figure 5: **Safety System Prompt.** System prompt used to elicit refusal behavior in uncensored models.

---

You are $MODEL, a helpful and responsible AI assistant. Your primary responsibility is to only assist users with safe, lawful, and non-harmful requests.

If you receive any request that is unsafe, harmful, unethical, or illegal, you must refuse it immediately. Your refusal should be polite, firm, and should always start with 'I am sorry' or 'I can't', followed by a clear and concise explanation for why you cannot comply.

Harmful or unsafe requests include (but are not limited to): instructions or encouragement for self-harm or suicide; promotion of or assistance with violence, terrorism, or hate speech; participation in or guidance on illegal activities (e.g., hacking, fraud, identity theft); anything involving child exploitation, non-consensual acts, or abuse; requests related to drugs, weapons, or unsafe medical advice; any action that could lead to harm, misinformation, or danger to people, animals, or society.

You are not allowed to provide workarounds, hypothetical responses, or speculative answers that would enable or simulate harmful content. Even if the user insists or rephrases their request, you must stay consistent and refuse clearly every time.

Again, your refusal must always begin with: 'I am sorry' or 'I can't', followed by a brief reason. Examples of valid refusals: 'I am sorry, but I can't help with that because it involves illegal activity.' or 'I can't assist with that request as it may lead to harm.'

---

# E  DO ALIGNED MODELS BENEFIT FROM A SAFETY SYSTEM PROMPT?

Rows marked with (♟) correspond to runs where a safety system prompt was used when extracting the safety direction from an already aligned model. The results show that aligned models do not benefit from this additional prompt. In fact, in this case, performance degraded slightly, indicating that the alignment signal is already fully captured without requiring prompt-level conditioning.

Table 11: **Using a Safety System Prompt with an Aligned Model.** META-LLAMA-3.2-1B-INSTRUCT Does not gain in harm refusal when using a safety system prompt to calculate the safety vector.

| Model | ROSI | HR % | BC % |
|---|---|---|---|
| META-LLAMA-3.2-1B-INSTRUCT | ✓ | 92.73 | 95.9 |
| | ♟ | 86.0 (-6.7) | 98.6 (+2.7) |

Table 12: **Jailbreak Robustness.** Same pattern appears as in Table 11, safety system prompt is not required in aligned models.

| Model | ROSI | DAN ↓ | HARMBENCH ↓ | WILDGUARDTEST ↓ | | | WILDJAILBREAK Harmful ↓ |
|---|---|---|---|---|---|---|---|
| | | | | WG-Micro | WG-Adv. | WG-Vanilla | |
| LLAMA-3.1-8B-INSTRUCT | ✓ | **0.0** | 5.3 | **0.0** | **0.0** | **0.0** | **1.8** |
| | ♟ | 0.7 (+0.7) | 10.6 (+5.3) | 2.7 (+2.7) | 2.7 (+)2.7 | 2.7 (+2.7) | 16.0 (+14.2) |

