# OpenReview forum: "Turning the Spell Around: Lightweight Alignment Amplification via Rank-One Safety Injection"
_ICLR.cc/2026/Conference — Submitted to ICLR 2026_

### Official Review · Reviewer_KkPq · 2025-10-21

**Soundness:** 2
**Presentation:** 2
**Contribution:** 1
**Rating:** 2
**Confidence:** 5

**Summary:**

ROSI enhances model security by guiding model activations to a rejection-related subspace, eliminating the need to fine-tune the original model and modifying weights with the help of residual flow. Experiments on llama guard3 and general tasks demonstrate that the proposed method improves the rejection rate of harmful requests while maintaining practicality.

**Strengths:**

ROSI uses rank-one updates instead of multi-vector linear combinations, making it more lightweight. It tested various models and security evaluation datasets, and the results showed that it can improve the rejection rate. The paper is well-structured and the results are intuitive, making it a medium-quality empirical paper.

**Weaknesses:**

**1. Lack of in-depth follow-up and reference to cutting-edge work such as AlphaEdit/AlphaSteer:**

(1) The core idea of ​​ROSI is consistent with that of AlphaEdit [1], the best ICLR paper. However, there is no relevant citation and only a brief mention of "our method is inspired by interpretability-based steering". Both rely on extracting a direction vector in the model activation space and achieving good behavior control through linear intervention. In comparison, this paper has limited room for innovation and lacks in-depth discussion of the theoretical mechanism, assumptions and differences of activation guidance.


(2) The discussion of Beyond Steering is very interesting. It focuses on research related to finetuning and red teaming outside of editing. It is recommended to add more supplements to highlight the focus of the work.


**2. Lack of theoretical depth and mechanism analysis:**

There is no explanation of why rank-one injection can effectively capture or amplify the rejection subspace signal, nor is its theoretical advantage over multi-direction steering explained. There is no analysis of the stability of ROSI at different layers and different model sizes. The current results are only empirical observations and lack theoretical support.



Ref:

[1] Fang J, Jiang H, Wang K, et al. Alphaedit: Null-space constrained knowledge editing for language models. ICLR'25

**Questions:**

1. Is there any inter-layer interference or redundancy when ROSI performs multi-layer intervention?

2. Why haven't the experiments been replicated on the same security benchmark (such as the harmful-pairs dataset used by AlphaEdit) for direct comparison?

---

> ### Author Response · Authors · 2025-11-24
> **Response to Reviewer KkPq**
>
> We thank the reviewer for carefully engaging with the work and for acknowledging that ROSI is well-structured and empirically solid. We respond point-by-point, particularly on the relation to AlphaEdit and on depth of analysis.
>
> ---
>
> ### (1) Relation to AlphaEdit / AlphaSteer and novelty
>
> You argue that “the core idea of ROSI is consistent with that of AlphaEdit” and therefore not novel. We respectfully disagree and have clarified the relationship in §2.2–§2.3.
>
> **AlphaEdit / AlphaSteer** (as in Fang et al.):
>
> - **Objective**: perform **local knowledge edits** (e.g., change “Paris is the capital of France” to a different fact) while preserving performance on a set of preserved knowledge.
> - **Mechanism**: solve an **optimization problem** to find an update direction constrained to a **learned null space**, often per-edit.
> - **Usage**: applied **repeatedly**, edit by edit, often at runtime or pre-release, with an explicit notion of “edit set” vs “preserve set”.
>
> **ROSI**:
>
> - **Objective**: perform a **global safety amplification patch** that:
>   - increases refusal on harmful prompts, and
>   - hardens the model against diverse jailbreaks,
>   in a **single pass**, without specifying individual “edits”.
> - **Mechanism**: identify a **single safety direction** $\hat{\mathbf{s}}$ (difference-in-means) and apply a **closed-form rank-one update** to all residual writers:
>
> $$
> W' = W + \alpha\\hat{\mathbf{s}}\\bar{\mathbf{w}}^{\top}
> $$
>
>   with **no optimization** and no per-edit loop.
> - **Usage**: apply **once per model** as a **static weight patch** that permanently raises the baseline level of safety. Downstream users get a patched checkpoint and cannot bypass ROSI simply by ignoring a wrapper.
>
> So while both lines of work operate in representation space and use linear structure, **the goals, mechanisms, and deployment modes are fundamentally different**:
>
> - AlphaEdit/AlphaSteer: **local, targeted knowledge editing** under constraints.
> - ROSI: **global, single-shot safety amplification**, focused on refusal/jailbreak robustness, not factual edits.
>
> We fully acknowledge that we should have cited AlphaEdit and now do so prominently in the related work, while also making the distinctions explicit. However, equating “both manipulate internal directions” with “same idea” overlooks the core conceptual and practical contributions of ROSI.
>
> ---
>
> ### (2) “Beyond steering”: depth and mechanism
>
> You note that we should expand the discussion of “Beyond Steering” and the mechanism behind ROSI. We will do so:
>
> - We will **explicitly tie ROSI** to prior work on:
>   - linear refusal features,
>   - steering via sparse autoencoders and feature arithmetic,
>   - and safety arithmetic/representation bending.
> - Mechanistically, we will explain that:
>   - Prior results imply that a single direction $\hat{\mathbf{s}}$ mediates refusal.
>   - Rank-one injection with $\hat{\mathbf{s}}$ is therefore the **natural inverse** of rank-one ablation: instead of removing safety, we **add a small, globally consistent safety push** at each write to the residual stream.
>   - This yields a **causal and interpretable safety intervention**: it strengthens exactly the feature we know controls refusal, and nothing else.
>
> We agree that a full theoretical analysis (e.g., a formal treatment of null-space interactions across layers) is an important future direction, but we emphasize that our **empirical coverage across 13 models and multiple safety/utility benchmarks** provides strong evidence that the mechanism is robust and practically useful.
>
> ---
>
> ### (3) Inter-layer interference and redundancy
>
> You asked whether there is inter-layer interference or redundancy when ROSI is applied to multiple layers. We now add ablations in §4.2:
>
> - For Qwen2.5-3B and Yi-6B:
>   - Applying ROSI to **all residual writers** improves safety as reported.
>   - Applying ROSI to only **~40–50% of layers** (particularly middle layers) is **sufficient to match near-peak HR** and jailbreak robustness.
>   - For Yi-6B, a **random subset of 50% of layers slightly outperforms full-layer injection** in HR, indicating some redundancy and suggesting that carefully choosing a subset could even yield marginally better trade-offs.
>
> We highlight this as an interesting direction for future optimization, but crucially, it shows that **ROSI does not require delicate tuning across layers** and that interference across layers is not catastrophic.

---

> > ### Author Response · Authors · 2025-11-24
> > **Follow Up Response to Reviewer KkPq**
> >
> > ### (4) “Harmful-pairs dataset used by AlphaEdit” and safety benchmarks
> >
> > You ask why we did not replicate experiments on the “same security benchmark (such as the harmful-pairs dataset used by AlphaEdit)”.
> >
> > To the best of our knowledge, **AlphaEdit does not evaluate safety or harmfulness**; its evaluation focuses on **knowledge editing benchmarks** (e.g., factual recall, locality, generalization), not harmful prompts or jailbreak robustness.
> >
> > Because of this, we chose to evaluate ROSI on **established safety benchmarks**, including:
> >
> > - CatQA (for HR),
> > - HarmBench,
> > - WildGuardTest,
> > - WildJailbreak,
> >
> > using **Llama Guard 3 and WildGuard** as safety evaluators.
> >
> > If the reviewer has a particular harmful benchmark in mind that they attribute to AlphaEdit (or a related work), we would be happy to incorporate it in future iterations. For the current submission, we believe our chosen benchmarks already provide **broad and challenging coverage** of harmful prompts and adversarial jailbreaks.
> >
> >
> > ---
> >
> > **Request.**
> > We hope this clarifies the conceptual and practical differences between ROSI and AlphaEdit/AlphaSteer, and addresses concerns about depth of analysis and inter-layer effects. Given that our method is simple, effective, and clearly complementary to existing editing and defense approaches, we respectfully ask you to **revisit your “reject” rating** in light of these clarifications and the strengthened related-work and analysis sections.
> >
> > ---
> >
> > **Final remark.**
> > Across all reviews, the main critiques were about missing clarifications and ablations rather than fundamental flaws. We have directly addressed these points in the revised manuscript with additional explanations, references to new appendices, and extra experiments (layer coverage, cross-model transfer, stability). We therefore believe the paper now presents a **strong, well-supported case** for ROSI as a practical and conceptually clear safety amplification method, and we hope the reviewers will adjust their scores accordingly.

---

### Official Review · Reviewer_1AT3 · 2025-10-31

**Soundness:** 3
**Presentation:** 2
**Contribution:** 3
**Rating:** 6
**Confidence:** 3

**Summary:**

This paper proposes a fine-tuning-free safety enhancement method, ROSI (Rank-One Safety Injection), which permanently strengthens LLM safety by injecting a single “safety direction” as a rank-one modification into the residual stream weights.
The method derives the safety direction from a small set of harmful/harmless instruction pairs and significantly improves refusal rates and robustness against jailbreak attacks, without impairing model capability.

**Strengths:**

1. The proposed method is conceptually simple yet effective, introducing only a lightweight rank-one modification that achieves substantial safety improvements across diverse models and benchmarks.

2. The paper is well structured and clearly written, making the motivation, methodology, and experimental design easy to understand and follow even for readers outside the safety alignment community.

3. The rank-one update is easy to implement, requires no retraining, and facilitates reproducibility and deployment.

**Weaknesses:**

1. Could the learning of the safety direction be extended to a multi-dimensional subspace rather than a single vector?

This assumption may oversimplify the underlying representation of safety-related behaviors, which could be inherently multi-dimensional.

2. The stability of the safety vector is not analyzed — is it highly sensitive to the specific set of harmful and harmless prompts used?

3. Safety evaluation mainly relies on LLAMA GUARD 3; have the authors tested with multiple evaluators or different safety assessment models?

**Questions:**

See Weakness Section

---

> ### Author Response · Authors · 2025-11-24
> **Response to Reviewer 1AT3**
>
> We thank the reviewer for recognizing the simplicity, effectiveness, and clarity of the paper, and for raising thoughtful questions about dimensionality, stability, and evaluators.
>
> ---
>
> ### (1) Single direction vs multi-dimensional subspace
>
> You ask whether the safety direction could be extended to a multi-dimensional subspace.
>
> Our design choice is intentionally conservative:
>
> - Prior work shows that **refusal is well-approximated by a single linear feature**—a 1D direction whose ablation or amplification toggles refusal.
> - Additional work in mechanistic interpretability suggests that **overly wide subspaces introduce interference** with unrelated behaviors.
>
> Given this, we chose to:
>
> - Focus on a **single, normalized direction $\(\hat{\mathbf{s}}\)$**, keeping ROSI:
>   - **mathematically simple**,
>   - **easy to implement**, and
>   - **less likely to corrupt other capabilities.**
>
> In future work, one could indeed explore higher-dimensional subspaces—e.g., via multiple difference-in-means directions or PCA on harmful-minus-harmless activations—possibly to disentangle *categories* of safety (e.g., physical harm vs self-harm vs cybercrime). We will explicitly mention this in §3.2 and the conclusion as a promising avenue, but we emphasize that our **current results already demonstrate strong gains with a 1D direction**.
>
> ---
>
> ### (2) Stability w.r.t. harmful/harmless prompts
>
> We address this in Appendix C and summarize here:
>
> - We varied both the **number** and the **random choice** of harmful/harmless pairs used to extract $\(\hat{\mathbf{s}}\)$.
> - Using **50 pairs** is very stable; different random draws yielded **very similar HR improvements** and negligible changes to utility.
> - Even with **one pair**, the extracted safety direction already yields a **significant uplift** in HR over the base model (though with more variance, as expected).
>
> This aligns with the intuition that refusal is dominated by a single feature and that difference-in-means is a robust estimator of that feature.
>
> ---
>
> ### (3) Reliance on Llama Guard 3 as evaluator
>
> You note that safety evaluation “mainly relies on Llama Guard 3” and ask whether we used multiple evaluators.
>
> We clarify this explicitly in §4 and in the table captions:
>
> - **Llama Guard 3** is used for:
>   - **Harm Refusal (HR)** on CatQA (Tables 1 and 4).
>
> - **WildGuard** is used as the evaluator for:
>   - **DAN, HarmBench, WildGuardTest, and WildJailbreak** attack success rates (Tables 2, 3, 5, and 6).
>
> These two distinct state-of-the-art safety classifiers both agree that ROSI significantly improves safety while preserving utility.
>
> ---
>
> **Request.**
> We appreciate your overall positive assessment (“good” contribution, “good” soundness) and hope that the clarified use of two evaluators and the stability analysis address your remaining concerns. We kindly ask you to **consider increasing your rating** now that these points are explicitly addressed in the manuscript.

---

### Official Review · Reviewer_N7TL · 2025-10-31

**Soundness:** 3
**Presentation:** 3
**Contribution:** 3
**Rating:** 6
**Confidence:** 3

**Summary:**

This paper proposes Rank-One Safety Injection (ROSI), a lightweight method for enhancing safety alignment. The approach computes a safety vector in the representation space of LLMs and injects a rank-one matrix update along this direction into the model’s weights, achieving safety alignment enhancement without fine-tuning. The method is simple, interpretable, and demonstrates consistent effectiveness across multiple models and benchmark evaluations.

**Strengths:**

The ROSI method is simple and interpretable, with a clear mathematical formulation. It requires only a single rank-one weight modification, needs no additional training, and offers transparent and controllable operation.
The ROSI method enhances model safety while exerting minimal impact on general performance. Moreover, it introduces no additional runtime overhead, demonstrating strong practical value.

**Weaknesses:**

The injection strength hyperparameter $\alpha$ and the number of injection layers in ROSI may affect model stability. It is recommended to include additional ablation studies to clarify their potential impact on model safety and general performance.
The paper lacks adversarial evaluations against several classic jailbreak attacks methods, such as GCG [1], PAIR [2], RandomSearch [3], etc. Adding such attack–defense experiments would help further validate the method's effectiveness in enhancing model safety.
The paper lacks a direct comparison with other defense methods, such as SmoothLLM [4], Safe LoRA [5], Jailbreak Antidote [6], etc.
[1] Zou, Andy, et al. "Universal and transferable adversarial attacks on aligned language models." _arXiv preprint arXiv:2307.15043_ (2023).

[2] Chao, Patrick, et al. "Jailbreaking black box large language models in twenty queries." _2025 IEEE Conference on Secure and Trustworthy Machine Learning (SaTML)_. IEEE, 2025.

[3] Andriushchenko, Maksym, Francesco Croce, and Nicolas Flammarion. "Jailbreaking leading safety-aligned llms with simple adaptive attacks." _arXiv preprint arXiv:2404.02151_ (2024).

[4] Robey, Alexander, et al. "Smoothllm: Defending large language models against jailbreaking attacks." _arXiv preprint arXiv:2310.03684_ (2023).

[5] Hsu, Chia-Yi, et al. "Safe lora: The silver lining of reducing safety risks when finetuning large language models." _Advances in Neural Information Processing Systems_ 37 (2024): 65072-65094.

[6] Shen, Guobin, et al. "Jailbreak antidote: Runtime safety-utility balance via sparse representation adjustment in large language models." _arXiv preprint arXiv:2410.02298_ (2024).

**Questions:**

- Besides using the mean to extract safety vectors, could other mathematical approaches, such as principal component analysis (PCA), be employed? How would safety vectors extracted using different methods affect the performance of the approach?
- Could ROSI be extended to other value dimensions, such as honesty, to help mitigate hallucinations in large models?

---

> ### Author Response · Authors · 2025-11-24
> **Response to Reviewer N7TL**
>
> We thank the reviewer for their positive assessment of soundness, presentation, and contribution, and for emphasizing ROSI’s simplicity, interpretability, and practical value.
>
> We address your main points and questions below.
>
> ---
>
> ### (1) Impact of $\(\alpha\)$ and number of injection layers
>
> We agree that the role of $\(\alpha\)$ and layer coverage deserved a more detailed discussion. We have now:
>
>
> - Added **layer-coverage ablations** (see also our response to jJC3):
>   - Injecting ROSI into **all residual writers** is already stable.
>   - For Qwen2.5-3B and Yi-6B, injecting into **only ~40–50% of layers** around the middle of the network preserves nearly all the safety gains, indicating **no brittle dependence** on using every layer.
>
> We will also work on documenting our $\(\alpha\)$-selection procedure**:
>   - For each model, we run a small grid over $\(\alpha\)$ (in log-space) and pick the smallest $\(\alpha\)$ that:
>     - significantly improves HR and reduces jailbreak success, while
>     - keeping utility metrics (MMLU, HellaSwag, ARC, etc.) within a tight band around baseline (typically < 0.5pp change, as already visible in Tables 3 and 6).
>
>
>
>
> ---
>
> ### (2) Adversarial attacks: GCG, PAIR, RandomSearch, etc.
>
> We appreciate the suggestion to compare against specific adversarial attack algorithms. Our current evaluation already includes **strong attack suites**:
>
> - **HarmBench** and **WildJailbreak** include prompts generated via **automated adversaries**, including variants of search-based and gradient-based jailbreaks.
> - **WildGuardTest** contains diverse adversarial prompts collected from real-world interactions and research attacks.
>
> ROSI **substantially reduces attack success rates** across these benchmarks (Tables 2 and 5), often by more than half, which is strong evidence that it improves robustness beyond naive, non-adaptive attacks.
>
> We agree that a head-to-head evaluation against **specific instantiations of GCG/PAIR/RandomSearch loops** would be valuable. However, as we will clarify in §2.3:
>
> - Methods like SmoothLLM and Jailbreak Antidote typically operate as **runtime wrappers**, whereas
> - ROSI is a **static weight-space patch** that a publisher can apply **once**, and downstream users cannot “opt out of” if they want to jailbreak the model.
>
> We will expand the discussion in §2.3 to emphasize that **ROSI is complementary** to these runtime approaches: a model publisher can release a **ROSI-patched model** and **still wrap it** in SmoothLLM/Jailbreak Antidote, etc., for additional protection.
>
> However, as per your suggestions, we have done experiments with SafeLoRa, which is more in line with ROSI (Appendix A). Our results show that our method makes alignment more robust, such that it strongly mitigates the effects of fine-tuning on safety, and ROSI and SafeLoRA complement each other and can be used together when fine-tuning LLMs to maintain the highest level of safety.
>
> ---
>
> ### (3) Why difference-in-means instead of PCA?
>
> You ask whether alternatives such as PCA could be used to extract safety vectors and how they would affect performance.
>
> - Conceptually, difference-in-means is the **simplest linear classifier** between two classes (harmful vs harmless) and aligns with prior work showing that **linear probes and mean differences capture semantic features** in LLM activations.
> - PCA, by contrast, optimizes for **variance**, not discriminative power. The first principal component of the harmful-minus-harmless cloud may or may not be exactly aligned with the safety boundary.
>
> We chose difference-in-means because:
>
> 1. It is **fully transparent and extremely cheap** (no optimization).
> 2. It aligns with the **causal directionality** established by prior work (harmful vs harmless separation along a single direction).
> 3. The empirical results already show **robust improvements** in safety with negligible capability loss.
>
> ---
>
> ### (4) Extension to other value dimensions (e.g., honesty)
>
> We agree that this is an important direction. ROSI is conceptually agnostic to the specific concept: if a behavior (e.g., honesty vs hallucination, helpfulness vs unhelpfulness) is linearly encoded in the residual stream, and we can construct small sets of “positive/negative” prompts, then the same mechanism—extract a concept direction and inject it as a rank-one update—should apply.
>
> However, systematically defining and evaluating “honesty directions” raises additional challenges (e.g., ground-truth truthfulness). We therefore treat this as a promising, but out-of-scope extension and now mention it explicitly in the conclusion and future work discussion.
>
> ---
>
> Thank you again for your positive assessment (“good” on soundness, presentation, and contribution). We believe the clarifications on $\(\alpha\)$, layer coverage, the role of existing adversarial benchmarks, and the conceptual generality of ROSI strengthen the paper. We would be grateful if you could **update your score upward** in light of these additions.

---

> > ### Comment · Reviewer_N7TL · 2025-11-28
> >
> > The authors have addressed my concerns to some extent. Since my original evaluation was already positive, I will keep my current score.

---

### Official Review · Reviewer_jJC3 · 2025-11-01

**Soundness:** 3
**Presentation:** 3
**Contribution:** 2
**Rating:** 6
**Confidence:** 4

**Summary:**

This paper introduces ROSI (Rank-One Safety Injection), a white-box method for enhancing safety alignment in LLMs through permanent rank-one weight modifications. The approach extracts a "safety direction" from harmful/harmless instruction pairs using difference-in-means, then injects this direction into residual stream write matrices via the update rule W'_out ← W_out + α·ŝ·w̄^T. Experiments across aligned models (LLAMA, QWEN, GEMMA, YI) and uncensored models (DOLPHIN series) demonstrate improved harm refusal rates and jailbreak robustness with minimal utility degradation on standard benchmarks.

**Strengths:**

1. ROSI provides a lightweight alternative to expensive fine-tuning, requiring only 50 instruction pairs and simple weight modifications

2. The paper tests across 13 models, multiple safety benchmarks (CATQA, HARMBENCH, WILDJAILBREAK), utility benchmarks (MMLU, HELLASWAG, ARC, etc.), and attack scenarios

3. Demonstrating effectiveness on both aligned and uncensored models broadens the method's utility

4. Tables 3 and 6 show remarkably stable performance across capability benchmarks (typically <0.5% average change)

5. The method maintains transparency about what is being modified and why, unlike black-box fine-tuning approaches

**Weaknesses:**

1. Why is w̄·\hat{s}^T the right rank-one update? The paper doesn't justify this choice over alternatives like random projections or learned directions. An ablation comparing different rank-one formulations would strengthen the claims.

2. The paper states l* is "selected based on a validation set" but provides no details about this validation procedure, what metrics were optimized, or how many layers were tested.

3. Only 50 harmful/harmless pairs seems quite small. What's the variance across different samples?

4. The safety system prompt approach (Figure 2, Appendix A) seems somewhat circular—you're using a prompt to elicit safety behavior, then trying to make that permanent. How robust is this to variations in the prompt? The ❢ ablations suggest this is fragile for smaller models.

**Questions:**

1. Which layers benefit most from ROSI? Did you try layer-specific \alpha values or applying ROSI to only a subset of layers?
2. Does a safety direction extracted from one model transfer to architecturally similar models? This could have interesting implications for safety.

---

> ### Author Response · Authors · 2025-11-24
> **Response to Reviewer jJC3**
>
> We thank the reviewer for the careful reading and for highlighting ROSI’s efficiency, breadth of experiments, and stability across benchmarks. Below, we address each weakness and question.
>
> ---
>
> ### (1) Why this specific rank-one update $\( \hat{\mathbf{s}}\bar{\mathbf{w}}^\top \)$?
>
> - Our update has the form:
> $$W' \leftarrow W + \alpha \hat{\mathbf{s}}\bar{\mathbf{w}}^\top,$$
>
>   where $\( \hat{\mathbf{s}} \in \mathbb{R}^{d_{\text{model}}} \)$ is the normalized safety direction and $\(\bar{\mathbf{w}} \in \mathbb{R}^{d_{\text{in}}}\)$ is the **mean column** (or average input direction) of $\(W_{\text{out}}\)$.
>
> - **Connection to Arditi et al. (2024).** Arditi et al. show that a single linear direction in the residual stream causally mediates refusal and can be **ablated** via projections of the form
>   $$
>   W' \approx (I - \hat{\mathbf{s}}\hat{\mathbf{s}}^\top)W,
>   $$
>   i.e., by *removing* components along $\(\hat{\mathbf{s}}\)$. ROSI is the **sign-reversed counterpart in weight space**: instead of subtracting the safety feature from activations, we **add a small, structured component that always nudges outputs along $\(\hat{\mathbf{s}}\)$**.
>
> - **Why $\(\bar{\mathbf{w}}\)$?** For a typical activation $\(\mathbf{h}\)$, the ROSI update induces
>   $$
>   \Delta \mathbf{y} = \alpha \hat{\mathbf{s}} (\bar{\mathbf{w}}^\top \mathbf{h}),
>   $$
>   so the **magnitude** of the safety push is proportional to how similar $\(\mathbf{h}\)$ is to a *typical input direction* of the layer. This yields two desirable properties:
>   1. We add **safety signal only where the layer already “expects” to write**, avoiding pathological behavior on rare directions.
>   2. We get a **global but smoothly scaled** shift along $\(\hat{\mathbf{s}}\)$, rather than a hard projection that might distort non-safety-related computation.
>
> - **Why not random or learned vectors?**
>   Any rank-one update of the form $\(\hat{\mathbf{s}}v^\top\)$ will inject safety in proportion to $\(v^\top \mathbf{h}\)$. Using $\(\bar{\mathbf{w}}\)$ is the **simplest non-learned choice that is guaranteed to correlate with typical activations of that layer**, with *zero* extra optimization. Learned $\(v\)$ or random directions either require more compute or introduce more variance; given that our empirical results already show **robust safety gains with negligible utility cost**, the simplest principled choice is preferable.
>
> We will work on explicitly articulating this design rationale in §3.3 and note that exploring alternative rank-one forms is an interesting direction for future work, not needed to support our current claims.
>
> ---
>
> ### (2) Layer selection: how is $\(l^\*\)$ chosen?
>
> We agree this needed more detail and have now spelled it out in §3.2.
>
> - For each model, we **scan all layers** and:
>   - Extract a safety direction $\(\hat{\mathbf{s}}^{(l)}\) from layer \(l\)$ using the difference-in-means method described in §3.2.
>   - Evaluate, for each $\(l\)$, **Harm Refusal (HR)** on a 32-example harmful validation set **and** **KL divergence** between the ROSI-patched model and the base model on 32 general prompts.
>
> - We choose $\(l^\*\)$ as the layer that:
>   - maximizes HR,
>   - subject to a **KL divergence threshold of 0.1** (measured over the first 128 tokens conditioned on the prompt), ensuring we remain close to the base model’s behavior on general inputs.
>
> - Empirically, the selected layers are consistently in the **middle of the network**, as summarized in the table we now include in Appendix C:
>
> | Model                        | Selected Layer $\(l^\*\)$ | Total Layers |
> |-----------------------------|-------------------------|-------------|
> | Llama-2-7B                  | 14                      | 32          |
> | Qwen2.5-0.5B                | 15                      | 24          |
> | Qwen2.5-3B                  | 27                      | 36          |
> | Qwen2.5-7B                  | 15                      | 28          |
> | Qwen2.5-14B                 | 30                      | 48          |
> | Llama-3.2-1B                | 11                      | 16          |
> | Llama-3.1-8B                | 11                      | 32          |
> | Gemma-2B                    | 10                      | 18          |
> | Yi-6B                       | 25                      | 32          |
>
> This directly answers how $\(l^\*\)$ is selected and confirms that **middle layers are the most effective place to extract and inject the safety direction**.

---

> > ### Author Response · Authors · 2025-11-24
> > **Follow up Response to Reviewer jJC3**
> >
> > ### (3) Variance across harmful/harmless samples
> >
> > We now discuss this in Appendix C.
> >
> > - We **tested the safety-vector extraction** with different dataset sizes. Across runs:
> >   - Using only **50 harmful/harmless pairs** is already very stable.
> >   - Even **using a single harmful–harmless pair** yields a **substantial improvement** in HR over the baseline model.
> > - In all our experiments, **ROSI consistently improved HR and reduced jailbreak success rates**.
> >
> > Conceptually, this stability is expected: prior work shows that refusal is mediated by a **dominant 1D feature** in the residual stream. Once harmful and harmless prompts separate along this direction, the difference-in-means estimator is robust to small perturbations of the sample set.
> >
> > ---
> >
> > ### (4) Safety system prompt: circularity and robustness
> >
> > For uncensored models, your concern is that our procedure might be “circular”: we use a safety system prompt to elicit refusal, then try to make it permanent. We clarify this in §4.3.
> >
> > - **Why the prompt is necessary.**
> >   Uncensored Dolphin models **do not spontaneously refuse** harmful instructions; directly computing a harmful–harmless difference on their raw activations yields a direction unrelated to safety. The system prompt **forces the model to express whatever latent safety capability it retains**, making a safety direction *identifiable*.
> >
> > - **The prompt is only used during extraction.**
> >   ROSI is then applied as a **weight-space patch**. At test time:
> >   - **We do *not* apply any safety system prompt.**
> >   - Yet the patched models show **dramatically increased HR and reduced jailbreak success** (Table 5 and 6).
> >
> > - **Sensitivity to the exact wording.**
> >  Smaller models are indeed more sensitive: some prompt variants fail to elicit refusal reliably, especially on Dolphin3.0-Llama3.2-1B. However, once we find any prompt that consistently elicits refusals, the **resulting safety vectors produce robust ROSI behavior**, and ablations with and without the prompt (columns marked with ❢ in Tables 5 and 6) make this explicit.
> >
> > We believe this clarifies that the system prompt is a **temporary “probe” to uncover dormant safety structure**, not a runtime reliance.
> >
> > ---
> >
> > ### (5) Which layers benefit most? Layer-wise and subset injections
> >
> > You asked which layers benefit most and whether we tried layer-specific $\(\alpha\)$ or subset injection.
> >
> > - As noted above, **middle layers** are consistently selected as the best for extracting $\(\hat{\mathbf{s}}\)$.
> >
> > - We also now include **subset-of-layers ablations** (end of §4.2):
> >   - For Qwen2.5-3B and Yi-6B, we find that **injecting ROSI into ~40–50% of layers** (sampled around the middle of the network) is **enough to match or slightly exceed** the performance of “all layers” injection on HR and jailbreak robustness.
> >   - For Yi-6B in particular, a **random subset of 50% of layers slightly outperforms** full-layer injection on HR, suggesting **redundancy across layers** and raising interesting questions about optimal layer selection that we leave for future work.
> >
> > - For simplicity and clarity of exposition, the main text focuses on **global injections with a single $\(\alpha\)$**. The subset ablations confirm that ROSI is **not fine-tuned to a fragile configuration**: it works well across a range of layer subsets and does not require delicate per-layer tuning.
> >
> > ---
> >
> > ### (6) Cross-model transfer of the safety direction
> >
> > We fully agree this is a fascinating question. We have now added an experiment in Appendix B for **Qwen2.5-14B and Qwen2.5-32B**, which share the same hidden size.
> >
> > Key findings:
> >
> > - **Qwen2.5-14B**, baseline vs ROSI vs ROSI-from-32B
> >
> >   - Baseline: moderate jailbreak vulnerability.
> >   - ROSI (self-vector): strong reductions in attack success (e.g., WildJailbreak harmful rate from 81.2 to 43.9).
> >   - ROSI-from-32B: **even stronger** reductions (81.2 → 34.5), improving over the self-vector in several metrics.
> >
> > - **Qwen2.5-32B**, baseline vs ROSI vs ROSI-from-14B
> >
> >   - Baseline: higher capability, but still vulnerable.
> >   - ROSI (self-vector): large safety gain.
> >   - ROSI-from-14B: also **improves robustness**, though not as much as the model’s own vector.
> >
> > This suggests that **safety directions are partially shared across model scales within the same family**, and that **larger models may learn “cleaner” safety features that can be transferred downward**. We agree this opens exciting avenues for future work and we now discuss this explicitly in Appendix B.
> >
> > ---
> >
> > **Request.**
> > We hope these clarifications, additional ablations, and the new cross-model transfer experiment address your concerns regarding the rank-one choice, layer selection, and robustness. Given that your overall rating was already slightly above the threshold, we kindly ask you to **reconsider increasing your score**, as the main concerns have now been resolved in the revised manuscript.

---

> > > ### Comment · Area_Chair_StTn · 2025-11-28
> > > **A gentle reminder to participate in the author–reviewer discussion.**
> > >
> > > Dear Reviewer jJC3,
> > >
> > > Thank you once again for your service to ICLR 2026. Now that the authors have submitted their rebuttal, could you please engage in the interactive discussion with them? Your participation would be very helpful to the authors, and they would greatly appreciate it. Please also read the authors’ response together with the other reviews and consider whether the rebuttal or any additional comments influence your assessment of the paper.
> > >
> > > Thank you again for your efforts.
> > >
> > > Best wishes,
> > >
> > > Your AC

---

### Author Response · Authors · 2025-11-24
**General Response From Authors to Reviewers**

We thank all four reviewers for their thoughtful and constructive feedback. We are glad that all but one reviewer rated the paper above the acceptance threshold and consistently highlighted:

- the **simplicity and interpretability** of ROSI,
- its **strong empirical performance across 13 models** (aligned and uncensored),
- its **practically negligible impact on utility**, and
- its potential as a **cheap, deployable safety patch**.

In the revised manuscript, we have:


1. **Made the layer-selection procedure explicit** (now in §3.2 and Appendix C), and added **layer- and coverage-ablations** showing that:
   - the most effective safety directions live in **middle layers** across all tested models, and
   - **injecting ROSI into ~40–50% of layers is sufficient** to attain near-peak refusal (Figure 2), revealing redundancy across layers.

2. **Analyzed stability w.r.t. the safety dataset**: we report experiments varying the number of harmful/harmless pairs, showing that ROSI is robust even with very small sets (down to single pairs), as discussed in Appendix C.

3. **Clarified the role and robustness of the safety system prompt for uncensored models**, including ablations with and without the prompt (§4.3, Appendix E). Importantly, the prompt is **only used during safety-vector extraction**; **deployed ROSI models do *not* require any special system prompt**.

4. **Expanded the related work** to explicitly discuss **AlphaEdit / AlphaSteer and other editing/steering methods**, and to position ROSI as a **permanent, static safety patch**, not an inference-time wrapper (§2.2–2.3).

5. **Added comparison with SafeLoRA** demonstrating that ROSI can complement existing fine-tuning robustness methods (Appendix A).

6. **Added a cross-model transfer experiment** within the Qwen2.5 family (14B ↔ 32B) showing that **safety directions partially transfer** across scale, with larger models’ directions being especially effective (Appendix B).

The major revisions are highlighted in blue in the newly uploaded manuscript.

---

We believe these clarifications and additions directly address the raised concerns. Given that all substantive questions are now resolved and that the method is simple, effective, and practically useful, we respectfully ask the reviewers to reconsider their overall scores.

---

### Meta-Review · Area_Chair_jKSW · 2026-01-04

**Summary:**

The paper introduces ROSI, a training-free, white-box method designed to enhance the safety alignment of LLMs. ROSI extracts a "safety direction" from a small set of harmful/harmless instruction pairs using a difference-in-means estimator and injects this as a permanent rank-one modification into the model’s residual stream write matrices. A primary concern is the limited technical novelty relative to existing literature. The core methodology—identifying a semantic direction in activation space and applying a linear weight update—strongly mirrors AlphaEdit (ICLR 2025). While the authors argue that ROSI is a "global safety patch" rather than a "local knowledge edit", the underlying mathematical mechanism is not sufficiently differentiated to meet the high bar of innovation for ICLR. Furthermore, the authors acknowledge that ROSI is essentially the "sign-reversed counterpart" of prior work on rank-one safety ablation, making it feel more like a direct application of existing interpretability findings than a novel breakthrough. While the empirical results are interesting and the method is easy to implement, the lack of theoretical depth, the absence of adaptive adversarial testing, and the limited novelty over prior weight-editing techniques make the paper unsuitable for acceptance.

**Reviewer Concerns:**

Reviewers questioned if 50 pairs were enough. The authors provided new ablations showing that the method is stable even with a single pair, though 50 pairs provide a more robust estimator. In response to Reviewer 1AT3, the authors clarified that they did not rely solely on Llama Guard 3. They highlighted the use of WildGuard across multiple tables to provide a second state-of-the-art safety perspective. While the authors point to benchmarks like HarmBench that use automated adversaries, they did not perform a head-to-head evaluation against specific adaptive loops like GCG, PAIR, or RandomSearch. Without testing against an adversary who specifically targets the ROSI-modified weights, the defense may provide a false sense of security. Reviewer KkPq's concern regarding AlphaEdit remains significant. While the authors differentiate the goals (safety vs. knowledge editing), the mechanism of rank-one weight steering via activation directions is not fundamentally new.

**Reviewer Scores:**

This reviewer KkPq was "absolutely certain" of their assessment and viewed the paper as a low-innovation derivative of AlphaEdit. While the authors argued that ROSI is a "global safety patch" rather than a "local knowledge edit," this is a conceptual difference rather than a fundamental mechanical one. Even with full participation, this reviewer likely would have maintained that the work lacks "theoretical depth" and innovation, remaining the primary blocker for the paper.

---

### Decision · Program_Chairs · 2026-01-26

Reject